# A qualitative study exploring the social and environmental context of recently acquired HIV infection among men who have sex with men in South-East England

Annabelle Gourlay,[1,2] Julie Fox,[3] Mitzy Gafos,[1] Sarah Fidler,[4] Nneka Nwokolo,[5] Amanda Clarke,[6] Richard Gilson,[1] Chloe Orkin,[7] Simon Collins,[8] Kholoud Porter,[1,2] Graham Hart[1]

► Prepublication history and additional material are available. To view these files please visit the journal online (http://dx.doi.org/10.1136/bmjopen-2017-016494).

For numbered affiliations see end of article.

**Correspondence to**
Dr Annabelle Gourlay;
a.gourlay@ucl.ac.uk, g.hart@ucl.ac.uk

## ABSTRACT

**Objectives** A key UK public health priority is to reduce HIV incidence among gay and other men who have sex with men (MSM). This study aimed to explore the social and environmental context in which new HIV infections occurred among MSM in London and Brighton in 2015.

**Design** A qualitative descriptive study, comprising in-depth interviews, was carried out as a substudy to the UK Register of HIV Seroconverters cohort: an observational cohort of individuals whose date of HIV seroconversion was well estimated. An inductive thematic analysis was conducted in NVivo, guided by a socio-ecological framework.

**Setting** Participants were recruited from six HIV clinics in London and Brighton. Fieldwork was conducted between January and April 2015.

**Participants** All MSM eligible for the UK Register Seroconverter cohort (an HIV-positive antibody test result within 12 months of their last documented HIV-negative test or other laboratory evidence of HIV seroconversion) diagnosed within the past 12 months and aged ≥18 were eligible for the qualitative substudy. 21 MSM participated, aged 22–61 years and predominantly white.

**Results** A complex interplay of factors, operating at different levels, influenced risk behaviours and HIV acquisition. Participants saw risk as multi-factorial, but the relative importance of factors varied for each person. Individual psycho-social factors, including personal history, recent life stressors and mental health, enhanced vulnerability towards higher risk situations, while features of the social environment, such as chemsex and social media, and prevalent community beliefs regarding treatment and HIV normalisation, encouraged risk taking.

**Conclusions** Recently acquired HIV infection among MSM reflects a complex web of factors operating at different levels. These findings point to the need for multi-level interventions to reduce the risk of HIV acquisition among high-risk MSM in the UK and similar settings.

## Strengths and limitations of this study

► This study provides important insights into the social and environmental contexts for recent HIV infections among gay and other men who have sex with men (MSM) in the UK, where reducing HIV incidence remains a public health priority.

► A unique feature was the recruitment of participants known to have recently acquired HIV infection, maximising their ability to recall contextual information.

► The primary limitation is the generalisability of findings that may not extend to MSM in other areas or ethnic groups, given that participants were exclusively recruited from London/Brighton and most were white. Nonetheless, our findings are likely to apply to similar UK or international cities with large populations of MSM.

► Social desirability bias is a further possible limitation, despite many respondents disclosing sensitive personal issues.

## BACKGROUND

Gay men and other men who have sex with men (MSM) accounted for more than half of individuals newly diagnosed with HIV in the UK in 2015,[1] with evidence of a rising trend in HIV incidence in recent years and continued high rates of transmission.[2–4] Understanding the context in which new infections occur in this key population therefore remains a priority.[5]

Qualitative and quantitative research conducted in the early 2000s identified behavioural factors associated with the risk of HIV acquisition among MSM in the UK, including condomless sex and an increasing number of partners, as well as the use of poppers (nitrite inhalants) and psychological

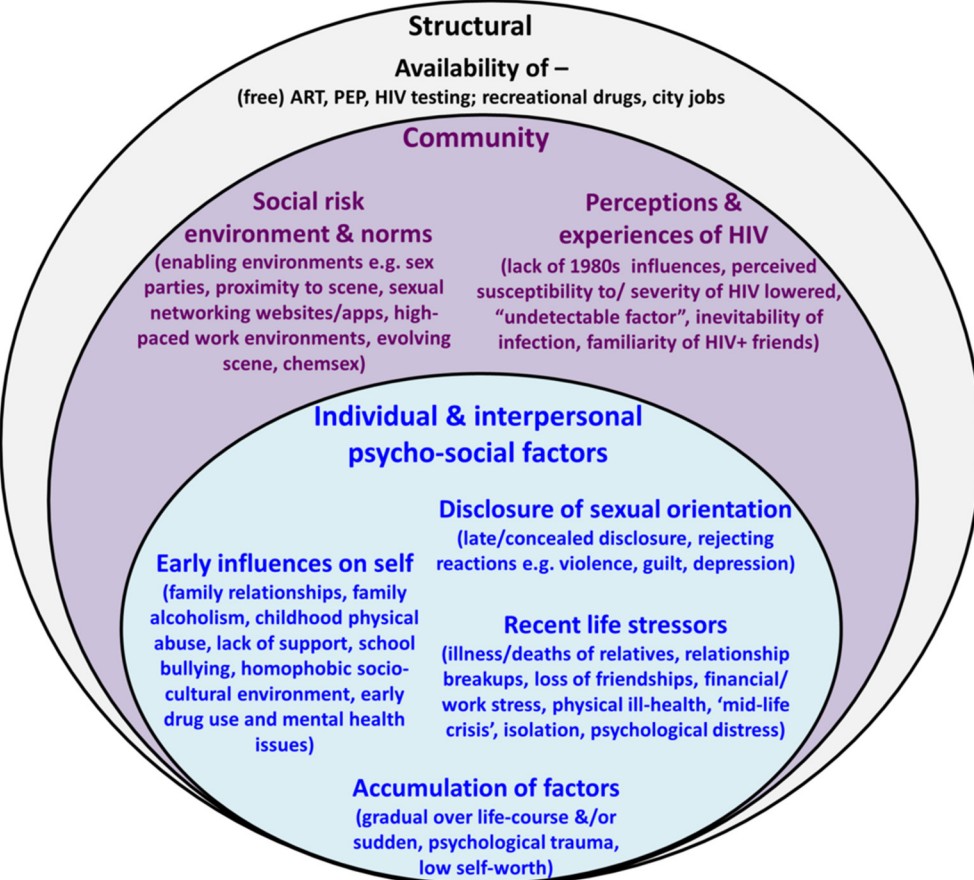

**Figure 1** Adapted socio-ecological framework of factors influencing risk of HIV acquisition among recently infected MSM in London and Brighton. This adapted socio-ecological framework[15 16] depicts individual-level and interpersonal-level psycho-social risk factors within the wider community and structural environments. Factors within and between each layer interact to influence risk behaviours and risk of HIV acquisition. ART, antiretroviral therapy; PEP, postexposure prophylaxis.

factors such as depression.[6 7] However, psycho-social factors were not explored or reported in detail, and their importance over a decade later remains unclear. More recent analyses using data from the PROUD trial identified ≥2 condomless sex partners in the last 90 days and having a bacterial rectal sexually transmitted infection (STI) in the previous year as the strongest risk factors for HIV acquisition.[8] However, qualitative insights into behavioural or social factors associated with HIV acquisition were not available.

Important changes have occurred over the last decade in HIV prevention and the social environment in which MSM interact. New medical interventions, such as the use of antiretroviral treatment (ART) as prevention,[9] pre-exposure prophylaxis (PrEP)[4] and the availability of HIV self-testing kits, are likely to influence perceptions of risk, risk behaviours and risk of HIV acquisition. For example, sexual behaviours might be influenced by the growing awareness that having an undetectable viral load (VL) dramatically reduces the risk of HIV transmission.[10] The context of sex between men has also changed dramatically, with the widespread availability of sexual networking through websites or mobile-phone-based applications ('apps') and greater availability of psychoactive drugs used for 'chemsex' (specifically mephedrone, γ-hydroxybutrate/γ-butyrolactone (GHB/

GBL) or crystal methamphetamine, taken intentionally to enhance sex). Evidence is emerging of high levels of engagement in these behaviours[11–13] with almost one in three HIV-positive MSM participating in a UK-based HIV clinic survey and 44% of HIV-negative MSM enrolled in the PROUD PrEP study reporting chemsex in the previous year.[12 14] However, the specific role of chemsex or social media and sexual networking in the acquisition of HIV or other STIs is less clear.

Relatively little is known about the social context of and behavioural responses to medical interventions, chemsex drugs or sexual networking media. Research is also lacking on how these factors, importantly, interrelate or interact with individual-level psycho-social factors, such as mental health, to influence HIV risk. Baral *et al* argued that, although epidemiological studies have traditionally focused on individual-level risk factors, it is essential to capture data that characterise multiple levels of HIV risk, including higher-order social and structural-level risks.[15] Levels of HIV risk and complex associations of factors at different levels can be appropriately investigated and depicted using socio-ecological frameworks, in which individuals and individual-level risks are conceptualised as part of the wider community and policy environment (figure 1).[15 16]

Through interviews with MSM who had recently acquired HIV infection, we set out to: (i) explore the social and environmental contexts in which HIV acquisition occurred among MSM in London and Brighton in 2015, (ii) investigate how these contexts influenced HIV risk and (iii) inform the design of public health interventions to help MSM reduce the risk of HIV acquisition.

## METHODS
### The UK register cohort study
We conducted a qualitative substudy to the UK Register of HIV Seroconverters: an observational cohort of individuals whose date of HIV seroconversion was well estimated.[17] Eligible individuals were ≥16 years of age and had an HIV-positive antibody test result within 12 months of their last documented HIV-negative test or other laboratory evidence of HIV seroconversion. Dates of seroconversion were estimated based on the latter, or the mid-point between last HIV-negative and first HIV-positive test result.

### Qualitative study design
The qualitative study was undertaken within the National Institute for Health Research Health Protection Research Unit (NIHR HPRU) in Blood Borne and Sexually Transmitted Infections (http://bbsti.hpru.nihr.ac.uk/).[18] We used a qualitative descriptive design[19] guided by socio-ecological theory,[15 16] employing in-depth interview data collection techniques and a thematic content analysis.

### Selection and recruitment of participants for interview
All MSM eligible for the UK Register, diagnosed within the past 12 months, aged ≥18 years and recruited from six centres in London and Brighton, South-East England, were eligible for the qualitative substudy. Centres were selected based on levels of recruitment to the cohort study and location, aiming to include areas with differing patient demographics. We restricted recruitment to these areas for logistical reasons and in view of their importance in the UK HIV epidemic, with over half of new HIV diagnoses among MSM in the UK made in London[2] and large MSM populations in both cities. HIV doctors undertook recruitment, explaining the study aims and providing information sheets. Only individuals who were willing and felt able to undertake the interview were recruited. Participants who gave their written consent to participate were contacted by an independent researcher (AG) or by a research nurse, as preferred, to schedule the interview.

Recruitment occurred between January and March 2015 until a minimum of 20 participants had been recruited and data saturation was reached on most themes. Thirty-six men were invited, five declined and six expressed interest but did not return to give consent. Of 25 who consented, 1 could not be contacted and 3 were unavailable for interview in the time-frame. Two men were subsequently excluded from the cohort, as their last negative test result could not be verified, but were included in the qualitative study.

### Data collection
In-depth interviews were conducted between February and April 2015 by an independent researcher (AG). Interviews took place in private rooms at each clinic and lasted 1–1.5 hours. Participants were advised they could decline to answer questions or stop the interview at any time.

The interviews followed a discussion guide (online supplement 1) including personal background, moving to London/Brighton and experiences of this transition, if applicable, life in recent years before HIV diagnosis, relationships and perceptions of the circumstances at the time of HIV infection. The guide was adapted during fieldwork to accommodate emerging themes.

Interviews were audio-recorded using a password-protected digital recorder, and summary notes were written after each interview. Recordings were transcribed verbatim and audio-files destroyed thereafter. Transcripts were password protected and stored on a secure computer network with restricted permissions. Transcripts were labelled with codes (eg, 'participant_A_date') and data reported anonymously.

### Data analysis
Summary notes informed preliminary analyses. A thematic analysis was then conducted on the interview transcripts using NVivo 11 software. Data were indexed and categorised to construct an analytic thematic framework. An inductive approach was used, further aided by topics on the discussion guide. A subsample (>10%) of transcripts was verified by a second researcher (MG) by constructing thematic maps and comparing themes, which showed a high level of consistency. Any discrepancies were resolved through discussions, resulting in some new categories being identified and added at this stage and some categories being refined. The revised framework was then reapplied to all transcripts. Data were subsequently summarised using thematic matrices[20] in Microsoft Excel to enable a comparison of themes within and across cases (interviews). The research team discussed themes at several stages of the fieldwork and analysis. A socio-ecological model was used to organise themes and data presentation (figure 1).[15 16] Interview quotations that best described the themes were selected for presentation, while also aiming to incorporate examples from most participants and all age groups.

### Ethics approvals
The qualitative study was approved by West Midlands, South Birmingham National Research Ethics Service, as an amendment to the UK Register protocol (reference 04/Q207155). Participants gave informed signed consent, including for the audio-recording of interviews.

**Table 1** Demographic characteristics of 21 participants

| Characteristic | Category | Participants (n) |
| --- | --- | --- |
| Age (years) | 20–29 | 6 |
| | 30–39 | 6 |
| | 40–49 | 5 |
| | ≥50 | 4 |
| Ethnicity | White | 19 |
| | BAME | 2 |
| Education | Secondary | 9 |
| | Higher education | 12 |
| Employment | Employed | 18 |
| | Unemployed | 3 |

BAME, black, Asian and minority ethnic.

## RESULTS

Interviews were completed with 21 participants, within 5 months, on average, of their HIV diagnosis and 6 months from their estimated date of HIV seroconversion. Participants were aged between 22 and 61 with a median age of 38 and were mostly white, well educated and employed (table 1).

We first report the psycho-social factors, described by the majority of participants, at the individual and interpersonal level that increased vulnerability to risk, including HIV acquisition. We then present themes at the community and structural level, including the social risk environment and community perceptions of HIV, which enabled risk taking. The relative importance of factors at each level and the way they came together to influence risk are illustrated in the final section and in figure 1.

## INDIVIDUAL AND INTERPERSONAL

Analysis of the in-depth interviews identified four main themes at the individual and interpersonal level. 'Early influences on self' encompasses childhood experiences associated with family—often characterised by poor relationships—school and participants' broader socio-cultural context growing up. 'Disclosure of sexual orientation' comprises the frequent rejecting reactions participants faced when disclosing their sexual orientation to others and the inner struggles and repercussions associated with initial concealment of sexual identity. 'Recent life stressors' are defined as distressing and sometimes severe events, such as deaths of relatives, experienced by many participants in the few years before their HIV diagnosis. The experience of multiple psycho-social issues, either as a series of sudden recent stressors or gradually since childhood, is captured in the fourth theme termed the 'accumulation of psycho-social risk factors'. We describe each theme in turn, drawing out their impacts on psychological well-being and then demonstrate how these intermediary psychological outcomes affected risk

behaviours, including risk of HIV infection, in the final subsection.

### Early influences on self

Experiences during childhood, within the family, school and the broader socio-cultural environment had early and long-lasting impacts on mental health, drug use and support structures. While some respondents reported strong relationships with family members, the majority described dysfunctional or superficial relationships with their parent(s), characterised by a lack of love or attention, abandonment, arguments, physical or verbal abuse, or parents/ siblings with alcohol or mental health issues. Many participants described how this led to low self-esteem, lack of confidence, insecurity, unhappiness, anxiety, depression or self-destructive behaviours.

*My father was…an alcoholic and he used to beat my mother and me…That may have had some impact on how destructive one is, and the fact I never had any unconditional love is something that I have struggled with in adulthood. (Aged 40–49)*

Family alcoholism occasionally influenced participants' own drug or alcohol habits, while early separation from family contributed to loneliness and early drug use for some participants. School-based bullying was recalled by several participants, with similar psychological repercussions.

*I: And were you able to identify any reasons behind that [anxiety]?*

*R: Yeah, lots and lots: mum, dad, school…I was far too shy a kid for boarding school. I'd get squished in the corner. (Aged 20–29)*

A few men were raised in cultures where gay men were commonly stigmatised or discriminated against, including highly religious contexts, abroad and/or in rural settings, which shaped self-perceptions regarding sexual identity and resulted in repressed sexuality or low self-esteem.

*It is quite difficult to understand in this day and age how prejudiced and narrow minded society was…where if I had been caught being gay…the school would have expelled me… never mind social views…So I was completely closeted and it became a huge frustration. (Aged 40–49)*

### Disclosure of sexual orientation

Most participants had disclosed their sexual orientation as teenagers or young adults. A few had initially resisted or concealed their sexual identity and discussed the psychological implications of self-denial.

*I was trying to change myself into a straight man…It didn't quite work out…I was basically on my own. My family didn't want to see me again…So the signs of depression were already there. (Aged 40–49)*

Several self-identified and came out as gay in their mid-twenties or thirties. For some, early concealment

and/or late disclosure were associated with resentment regarding missed opportunities, prompting a desire for sexual exploration.

*Growing up in an environment where you are getting to know yourself quite late, you get to…thinking about experiences and seeing other sexual stuff that you might not have needed to think about before because you are a bit behind…You know, am I missing out on stuff?* (Aged 40–49)

Participants had commonly experienced negative reactions to their disclosure. Several described 'traumatic' responses, including physical violence and rejection, which led to feelings of guilt or depression, frayed family relationships and weakened support networks.

*It [coming out] was quite traumatic at the time. My mum has never knowingly under-reacted to anything.* (Aged 40–49)

### Recent life stressors

Recent stressful events experienced prior to HIV diagnosis caused psychological distress for many participants. These encompassed severe illnesses or deaths of relatives, relationship break-ups, partner/ sexual violence, loss of friendships and bouts of physical ill health.

*I: Can you identify a particular trigger for that [depression]?*
*R: I think it was a consequence of events…It was…falling in love with someone who I couldn't be with…splitting up with my long term partner.* (Aged 40–49)

A number reported financial or work stress, including harassment, as key life stressors. Alcohol or drug use, as well as loneliness and isolation were common outcomes, associated with fragmentation of support networks.

*About 2013, that's when everything started to get sour at work…a lot of stress, being really unhappy…relying on drinking.* (Aged 30–39)

A few middle-aged men described a 'forty-something crisis', unfulfilled ambitions or wavering self-worth.

*I mean it probably was the perfect storm you know, they [drugs] got me at a time…mid-forties when I wasn't that secure, there were a few issues, I was looking for fun…it was an escape and it seemed at the time that it was…enjoyable.* (Aged 50–59)

### Accumulation of psycho-social risk factors

Several individuals were exposed to multiple psycho-social risk factors from a young age, which accumulated gradually.

*We have to go back to my childhood…I think everything [experiences, for example, childhood physical abuse, early concealment of sexual identity, depression, mother's recent death] adds a weight and it's been add and add and add to a point where I lost control…I decided to let myself go.* (Aged 40–49)

In other cases, or additionally, a series of destabilising, traumatic events occurred suddenly.

*The year before last it was like every other week something was happening or somebody was dying and my mum had a stroke, my dad had a heart attack, [partner's] dad was involved in a really bad car crash and nearly died and… then his auntie was diagnosed with motor neurone [disease] and it was like "anything else?"…It was just so much.* (Aged 30–39)

In both scenarios, individuals became overwhelmed, unable to cope, lost control or experienced psychological trauma, including low self-worth, loss of purpose, depression and suicidal thoughts.

*I was probably overwhelmed, you know. I wasn't in a stable place, I wasn't in a stable relationship, I wasn't stable financially and I had just suffered some pretty serious losses in terms of my immediate family. It was kind of all over the place really.* (Aged 40–49)

### Impacts on risk behaviour

Psychological impacts sometimes directly affected sexual behaviour, for example, sex sought for escapism or self-validation. One man, who said he never 'felt nurtured' by his parents explained:

*I always need validation from people…and that manifests itself in a sexual context.* (Aged 40–49)

Loneliness and a desire for intimacy also prompted condomless sex. For example, one man who described himself as previously 'very strict with sex practices' said:

*It was probably because of the breakup of my relationship I was just feeling a need to be close to people, and that often came out in me choosing to have sex, unsafe without a condom.* (Aged 30–39)

Some individuals deliberately put themselves at risk, for example, having condomless sex, including with HIV-positive partners, to self-harm or regain control, but did not indicate that this was a recent change in their behaviour.

*A lot of the difficulties I have…were about feeling controlled or not in control…so I knew what the risks were…but it was my choice, my decision [to have condomless sex with known HIV-positive partners].* (Aged 30–39)

Changes in attitudes to risk were also brought about by emotional trauma. A few participants described re-evaluating the potential costs of unsafe sex and the risk of HIV infection, relative to other more important life events, or justifying their risk taking during transient periods of psychological distress. Others cared so little for themselves that getting HIV seemed inconsequential.

*I didn't value my life…Because so much had happened and I'd been through so much in the past 3, 4, 5 years with… break ups and losing everything and emotional things and deaths and God knows what else, it almost becomes a bit "all*

*my life has just been so crap anyway what's the point, do I really care if I get it [HIV] anyway?".* (Aged 30–39)

## COMMUNITY AND STRUCTURAL

At the community and structural levels, two key themes emerged. 'Social risk environment and norms' captures the temptations of the evolving social scene, for example, sexual networking via social media and normalisation of risk-taking behaviours such as chemsex, interlaced with structural factors such as availability of recreational drugs, that exposed individuals to potential risks including HIV acquisition. The second theme, 'community perceptions and experiences of HIV', describes shifting community attitudes towards HIV, influenced by structural factors such as the availability of ART, postexposure prophylaxis (PEP) and HIV testing, and the impacts in terms of reduced risk perceptions and willingness to engage in risk behaviours.

### Social risk environment and norms

Participants were attracted to London and Brighton for the open-minded culture, freedom and social opportunities offered, although most had lived in these urban areas for a number of years. Almost all had socialised or met sex partners at saunas, clubs, private sex parties, 'chill-outs' (after-parties, typically involving drugs and/or group sex) or cruising grounds. For a few city workers, the intensity of partying was stimulated by challenging and high-paced work environments. Temptations of the scene were hard to resist for some men, while physical proximity to venues increased social opportunities and often exacerbated high-risk behaviours. Many participants talked about the excesses of partying on the scene and roughly half-talked about their experiences of weekend-long binges of drugs and sex.

*You go to Vauxhall [district of South London with a high-density of gay venues] on a Friday night as a gay man and don't come home until 5 days later. I think there is so much to tempt young people these days, I include myself in that.* (Aged 20–29)

Social media, including geo-locational sexual networking apps, also provided convenient access to multiple sexual partners for many participants, regardless of age. Social media were often used deliberately in pursuit of sex and/or drugs and widened sexual networks including with HIV-positive men.

*The introduction of gay apps just makes the sex so much more promiscuous…I think people get a rush from it. I probably did at the time. These apps…just make it so much more convenient. They had these code words for 'come round to ours and do this'…'have this drug'…'chill-outs'… It all escalates.* (Aged 20–29)

The gay scene was described as 'fun', 'vibrant' and 'exciting', but also 'reckless', 'dangerous' and 'abusive'.

Several men had experienced physical or sexual violence at saunas or nightclubs, sometimes associated with drug use, while cruising, or through meeting people via social media.

*Around that time there were two rapes on me…one happened in…a night club down the road…I went under on G [was under the influence of GHB/GBL]…and was attacked.* (Aged 20–29)

Several participants recalled changes in the gay scene, particularly the culture of drug use and sex. Changes were described in drug choices (eg, mephedrone, GBL/GHB, crystal methamphetamine), context (chemsex), modes of administration (normalisation of injecting) and increasing availability of drugs at lower costs.

*Drugs have changed…there are more choices…GHB, mephedrone…which I was quite scared of in the beginning… but then it's normalised in the gay scene and you just tend to do what other people do. Same thing goes for injecting. I mean these days it's not seen as so scary.* (Aged 40–49)

A few other participants discussed the role of social norms and networks, as well as the pursuit of pleasure, as reasons for getting into chemsex:

*There's more drug taking in my social circle than I realised. I did occasionally take them (other recreational drugs), but it was the sex thing…I think it was discovering how good the sex was…on drugs in terms of the duration and intensity.* (Aged 50–59)

Chemsex enhanced and extended sex, reportedly impaired judgement, decreased risk perceptions, and moved boundaries.

*I wasn't particularly stoned but you lose certain inhibitions and…I know there was a guy having sex with me, I was bottom, and I knew he had a condom to start off. But…I'm not quite sure he had a condom after.* (Aged 50–59)

### Community perceptions and experiences of HIV

Respondents, particularly middle-aged or older men, described shifting perceptions about HIV. Some felt that the absence of stark media campaigns and deaths from AIDS associated with the 1980s–1990s had reshaped attitudes towards HIV, reducing fears and risk perceptions. HIV was now frequently likened to having asthma or diabetes, was less concerning than hepatitis C and was no longer viewed as a death sentence.

*I think in London it's almost got to the point where people are not that concerned about it anymore. It's not looked at as a death sentence. I remember reading an article by a doctor, which I know a lot of gay people seem to have read…that he would rather have HIV than diabetes.* (Aged 30–39)

Living a normal, healthy life with HIV was often attributed to the availability of ART and good medical care. Believing that HIV is a manageable condition, reduced fears of HIV and knowing healthy HIV-positive men led

to complacency, with some men admitting their general denial or disregard of risks.

> *Everyone knows somebody positive now and knows that they're fit and healthy and they take a few pills a day… That's a huge factor in why so few people use protection anymore…because it has become a treatable illness…I think it changed everyone's risk calculations, because even if the worst did happen, it was no longer the worst. (Aged 20–29)*

Several participants said that they balanced risks by testing regularly for HIV and other STIs and/or that repeated negative results had encouraged further risk taking.

> *I get into the way of thinking, "oh it doesn't matter, you're going to die anyway one day…If you get sick you can take the pills" and it all becomes a bit blasé and a bit reckless. So it's taking a risk and then going to the clinic and finding out you're all clear. (Aged 50–53)*

Participants commonly associated an undetectable VL, and occasionally ART, with lower risk of HIV transmission, a few specifically recalling scientific evidence. Some participants, typically younger men, made conscious choices to engage in condomless sex, including with HIV-positive (usually undetectable) men, influenced by their awareness of transmission risks and/or availability of ART, beliefs that HIV was no longer the worst outcome or was inevitable considering the 'high prevalence of HIV in London'.

> *So we weren't always safe…I've read a lot…I think there was a French one [study] which was undetectable partner, negative partner, X number of however many tens of thousands of interactions, no transmissions. So, background reading kind of made me feel more comfortable with it [condomless sex with HIV-positive ex-partner]. (Aged 20–29)*

Another participant, who said he had always had condomless sex, highlighted how sex with HIV-positive undetectable men was sometimes perceived as less risky than with HIV-negative men:

> *I was going to try and minimise the risks…and so I slept mostly with undetectable guys. I tried to avoid negative guys like the plague…Everyone told me, and what the research told me too, was that they were…the only ones who were going to possibly infect. (Aged 20–29)*

A few participants mentioned PrEP spontaneously and had asked for it at their sexual health clinics but were not able to access it, while several had taken post-exposure prophylaxis (PEP) or discussed common perceptions of PEP as a backup option. Generally, many respondents thought that the availability of ART, PrEP, PEP or HIV tests, 'the undetectable factor', changing discourses around HIV or declining stigma had created a sense of apathy towards risks and susceptibility to HIV in the gay community. One man explained:

> *It's almost like they've been cured, "Oh yeah, I'm HIV-positive but I'm undetectable"…which lessens the risk.* (Aged 20–29)

## COMBINED EFFECTS AT THE INTERFACE OF INDIVIDUAL AND COMMUNITY LEVELS ON RISK

Our analysis revealed a layered complexity, with themes from both the individual and community/structural levels influencing participants' risk behaviours and risk of HIV infection. In this section, we describe and give examples of combinations of factors from multiple levels that individuals attributed to their HIV infection while also showing how the relative importance of contributory factors from individual and community/structural levels differed for each individual. We further illustrate how themes from each level could come together to heighten exposure to risk situations, for example, via the effects of impaired psychological well-being on sexual risk behaviour or elevated use of drugs and sexual networking apps, coupled with the lures of the social environment and exposure to high-risk contexts for sex. Changes in perceptions of risk and risk–benefit decisions around condomless sex were also brought about by individual-level psycho-social influences and experienced psychological trauma, which were compounded by changing community attitudes towards HIV and structural-level factors such as the availability of ART.

### Combinations of factors and perceived source of HIV infection

A minority of participants thought they could pinpoint their source of HIV infection, although several indicated that the point of infection was not important to them and mentioned that they did not like to dwell on this or blame anyone. Most men thought they had contracted HIV from a casual sexual encounter at a club, sauna or sex party or from someone they had met via social media. A few suspected their source of infection was a more regular partner, while a few cases of violence, for example, sexual coercion or being purposefully overdosed by someone, were possibly related to HIV infection. Some participants discussed changes over time in their sexual risk taking, for example, increasing frequency of sex or beginning to have condomless sex, which were occasionally rationalised in terms of recent stressors. However, a greater number of men did not think their sexual behaviour had changed, and some were surprised that they were not infected sooner.

Few participants thought that their HIV infection was the result of a single factor. More generally, participants thought that combinations of factors at both individual and community/structural levels contributed to risk behaviours and HIV infection. Within the combination of factors that contributed to HIV acquisition (figure 1), the relative importance of factors from each level differed for each person.

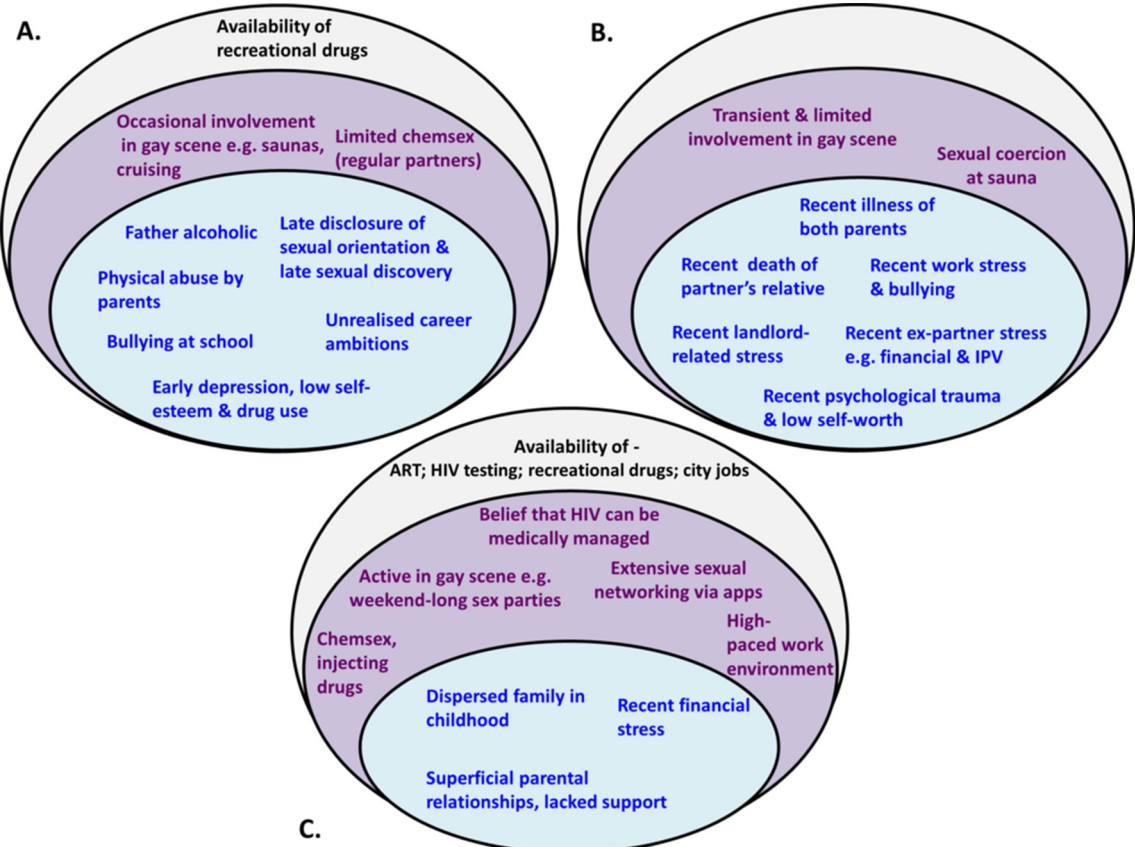

**Figure 2** Examples of combinations of factors, applying a socio-ecological framework, which contributed to risk behaviours and/or HIV acquisition for three participants. Individual and interpersonal psycho-social factors are depicted in the inner circle, factors at the community level are shown in the middle circle, and factors at the structural level are depicted in the outer circle. In all three cases, factors from all levels contributed to HIV risk, although their relative importance varied. Cases A and B both experienced a high density of individual-level psycho-social factors, which played a greater role than community or structural-level factors. These two cases also illustrate a contrasting balance of factors within the individual level, with case A having experienced psycho-social factors that accumulated from a young age, and case B having experienced only more recent stressors. In case C, individual-level psycho-social factors contributed to a lesser extent than community and structural-level factors. ART, antiretroviral therapy; IPV, intimate partner violence.

A few participants specifically attributed their risk of HIV to early psycho-social issues as well as the social environment. For example, one participant felt that his self-harming sexual behaviours stemmed from childhood and the violent relationship with his mother but also high-lighted the role of the 'abusive' environment, including gay saunas, and normalisation of drug use:

*I think with the sex, I think it's…environment, especially in South London. The increase of risk sex, chemsex, is becoming an epidemic, in my opinion. You hear of so many young gay men now who are positive…and through this lifestyle. It's very hedonistic, really nasty…I think, subsequently, living in South London has made me get HIV.* (Aged 20–29)

### Pathways to risk at the interface of individual and community levels

Some explained how psycho-social issues resulted in their elevated use of drugs or sexual networking apps, which in turn widened their exposure to contexts for higher risk sex such as sex parties and chemsex.

*I think the drugs make you want to do things like that [sex/apps]. Being unhappy at work, the incredible amount of stress I went through, all of that combined made me feel a bit lonely as well…So I started to rely a bit on Grindr.* (Aged 30–39)

Psychological issues and drug use were often mentioned in combination. For example, one participant identified the important factors in his HIV infection as:

*The drugs…but also depression because I didn't care about taking risks…I gave up.* (Aged 40–49)

Some participants who experienced psycho-social issues also identified ART and changing perceptions of HIV as factors that had consciously, or subconsciously, influenced their risk behaviours. Risk–benefit decisions regarding sexual practices were altered, influencing decisions to engage in condomless sex, including with HIV-positive men, or imbuing a more general disregard of risks.

*When we were young adults, the fear of God was put into us. If you got it [HIV] you died…Now it is manageable…you*

*could live a normal life…and you are thinking "Okay, I live in a city with very high prevalence of this disease. I am at a crossroads in my life in terms of what I want to experience sexually" …I think the trauma I have gone through has changed what I perceive as real risk in life… There are far more important things…So it was a combination of all those different aspects, I had come to the conclusion that if I did become HIV-positive it wouldn't be a big event in my life.* (Aged 40–49)

The environment also enabled sexual exploration, particularly for those who had initially contained their sexuality, for example, due to socio-cultural influences in their childhood. For a few participants, moving to London and fragmenting support networks had coincided with coming out, exposing them to new social and sexual norms including sex parties, group sex and chemsex.

*I was like a kid in a sweet shop. I wanted to experience everything, but I didn't necessarily have the rhetoric [support] around me to keep me safe.* (Aged 30–39)

### Relative importance of factors

In contrast to the weight attributed to psycho-social factors by participants in the examples above, other participants thought that the social environment played a more substantial role than psychological factors in the build up to their diagnosis. For example, one man emphasised that recent stress, such as his father's ill health, had less bearing on his 'dangerous lifestyle' and HIV infection than the draws of the social scene:

*Probably they [stressors] were a lesser extent than "I want to go out and have fun because I'm getting old and I'm not going to be able to do it in ten years' time", and that pull was probably stronger than any of the negative stuff about my life.* (Aged 50–59)

Some felt strongly that increasing use of sexual networking apps had played a major part in their own infection, through the ease of access to sexual partners, or in rising HIV infections in general, or that certain apps promoted promiscuity and irresponsibility. However, others reiterated that it was difficult to distinguish the primary factor amidst a range of social environmental influences.

*The sex and the drugs and the apps all intertwined simultaneously and I can't really say which one led to the other.* (Aged 20–29)

Figure 2 gives worked examples of three contrasting cases, using the socio-ecological model, demonstrating the complexity of influences within and across multiple levels for each individual in terms of their risk of HIV acquisition. These examples also highlight the differences in relative importance of factors from each level, where for some men, individual-level psycho-social factors played a greater role in their infection compared with community-level or structural-level factors (eg, cases A

and B), while for others, community or structural-level factors contributed to a greater extent (eg, case C). Within the individual level, there was typically a distinction in the balance of psycho-social influences towards those experienced early (eg, case A) or more recently (eg, case B), which in either case may accumulate to influence HIV risk.

### DISCUSSION

This qualitative study among recently infected MSM revealed a complex interplay of factors at the level of individuals and their community that influence risk behaviours and HIV acquisition. The relative importance of factors at each level varied for each person, with individual psycho-social factors enhancing vulnerability towards sexual risk situations and features of the social risk environment and prevalent community beliefs encouraging risk taking. A synergy of factors at the interface of these levels was ultimately important in affecting risk. Our findings reveal the broader context around new HIV infections in this population.

Psycho-social issues including early experiences of physical abuse, social rejection, isolation, drug use, anxiety or depression were common and heightened vulnerability to risk. Mental health conditions including major depression, anxiety disorders, suicidal tendencies, psychological distress and low self-esteem have been consistently reported at higher levels in MSM versus heterosexual men in high-income settings.[21 22] Several studies, predominantly American and including MSM populations, have also linked psycho-social issues such as depression, partner violence and childhood sexual abuse with condomless sex or risk of HIV acquisition.[6 23–26]

Notably, we found that a considerable number of participants had experienced multiple psycho-social issues over their life course and/or a sudden series of traumatic events in the few years preceding their HIV diagnosis. These severely damaged their psychological well-being and appear important in placing some individuals at particularly high risk of engaging in unsafe sex or acquiring HIV. The magnified effect of experiencing multiple psycho-social issues is consistent with 'syndemic' theory, proposed initially to describe concurrent and mutually exacerbating epidemics, including poverty, racism, violence, substance use and HIV, in ethnic minority populations in the USA.[27] This concept was later extended to urban gay men, with depression, partner violence, suicidality, substance abuse and HIV identified as inter-related reinforcing epidemics.[22 25] Another US-based longitudinal study has since provided compelling evidence that the accumulation of psycho-social conditions predicts HIV-related risk behaviours and HIV seroconversion among MSM.[24]

Environments that normalise risk taking, which in itself is a natural part of sexual discovery, can increase the likelihood of HIV acquisition, particularly for men whose individual-level experiences and other circumstances reduce their prioritising of sexual health. Chemsex was

frequently reported in our sample and facilitated exposure to potential risks, for example, through impaired judgement. This concurs with the findings of another qualitative study conducted in South London, which demonstrated that chemsex can carry more risk than drug-free sex, typically through longer duration and a greater number of partners.[28] Underlying psycho-social factors, early-onset or more recent stressors, were occasionally related to engagement in chemsex in our study, although social norms sometimes appeared to be stronger influencing factors.

Sexual networking apps were commonly an accessory to drugs and sex, making it easy to meet partners and connect with networks where risk of HIV transmission might be higher. Our findings complement those of a recent cross-sectional study among young MSM in Britain and Ireland, suggesting that higher odds of high-risk condomless sex were associated with, though not necessarily caused by, longer use of these technologies.[13] However, our results highlight the difficulty in disentangling the role of sexual networking apps from other social environmental influences, including chemsex and other drug use, or underlying psycho-social influences.

Influenced by structural factors including the availability of ART, HIV was widely perceived among the men interviewed to be preventable and manageable, shaping attitudes to risk and behavioural norms that, in turn, played a role in HIV acquisition. Some men had developed risk-reduction strategies accordingly, while reduced perceptions of risk also led more generally to complacency in terms of sexual practices. Findings to date on this topic have been mixed,[29–33] with causality difficult to infer. However, our findings support those of more recent studies,[10 34] suggesting that optimism about HIV/treatment is playing an increasingly important role in sexual risk taking.

The main strength of our study lies in the richness of data collected, covering a broad range of recurrent themes. Furthermore, interviews took place relatively recently after infection when recall was likely to be good. Nonetheless, details of participants' likely source of infection and related events may, naturally, be limited in accuracy. Many respondents disclosed sensitive personal issues, although we cannot rule out the possibility of social desirability bias. Participants were exclusively recruited from London/Brighton and most were white, so the generalisability of findings to MSM in other areas, or to black, Asian and other ethnic minorities, may be limited. However, we focused on a key population of public health importance in the UK.[2] It is also likely that our findings are applicable to MSM in other large cities, in the UK or similar international contexts, with sizeable MSM populations.

Our findings highlight the need for expanded access to packages of interventions that address exposures at multiple levels, if we are to limit new infections in this population. Examples already exist of multi-level or multi-faceted interventions[35] that, moving forwards, must be prioritised, particularly in the context of diminishing funding for sexual health and fragmentation of health service commissioning. Key components of a multi-level intervention would include, ideally, at the individual level: improved clinical assessments to identify high-risk HIV-negative MSM who may benefit from enhanced support, including tailored support for young MSM, given the prominence of psycho-social issues, some of which emerged during adolescence; providing timely and flexible support, for example, on-site in STI clinics and available immediately after testing, or via virtual web-based technologies; and outreach activities to link vulnerable MSM to peer or community-based support groups, considering the deleterious effects of isolation and lack of social support. To address community-level risk factors, programmes should incorporate the provision of further information and education about chemsex, ideally including on-site advice in STI clinics. Training of health professionals on the social context around chemsex and sexual norms is essential, to enable culturally competent discussions with MSM about their sexual behaviour, including engagement in chemsex, their risk-reduction strategies, and potential risks. In addition, sexual networking platforms are well placed to deliver targeted interventions, including chemsex awareness and counselling options, while peer-led initiatives could assist in addressing complacency associated with shifting perceptions of HIV and ART, for example, by raising awareness of the social and psychological implications of an HIV diagnosis. At the structural level, our findings support policies that include access to PrEP, in combination with adherence support and promotion of frequent STI testing, for periods in which MSM find themselves at elevated risk due to compounded psycho-social life stressors.

In this paper, we have shown that recently acquired HIV infection among MSM reflects a complex web of factors operating at different levels. Individuals who experienced multiple stressors, gradually over the life course or more suddenly, were especially vulnerable to HIV and being drawn into sexual risk situations, while the social environment created a context that enabled risk of HIV infection. While individuals were exposed to different sets of contributing circumstances, the interface of factors at each level was important in influencing risk behaviours and HIV acquisition. The circumstances surrounding HIV acquisition are complex and therefore require multi-level interventions that address individual and interpersonal psycho-social, community and structural-level risk factors.

**Author affiliations**

[1]Faculty of Population Health Sciences, University College London, London, UK
[2]National Institute for Health Research Health Protection Research Unit in Blood Borne and Sexually Transmitted Infections, London, UK
[3]Guys and St Thomas' NHS Trust/ King's College London, London, UK
[4]Department of Medicine, Imperial College London, London, UK
[5]56 Dean Street, Department of HIV/GUM, Chelsea and Westminster Hospital NHS Foundation Trust, London, UK

[6]Elton John Centre, Royal Sussex County Hospital, Brighton and Sussex University Hospital, Brighton, UK
[7]Ambrose King Centre, Royal London Hospital, Barts Health NHS Trust, London, UK
[8]HIV i-Base, London, UK

**Acknowledgements** We would like to thank all the participants who took part in the interviews, as well as the clinic staff who assisted with recruitment. This report is an independent research by the National Institute for Health Research. The research was funded by NIHR HPRU in Blood Borne and Sexually Transmitted Infections at University College London in partnership with Public Health England (PHE) and in collaboration with the London School of Hygiene and Tropical Medicine. The views expressed in this publication are those of the author(s) and not necessarily those of the National Health Service, the National Institute for Health Research, the Department of Health or Public Health England. Members of the Steering Committee of the NIHR HPRU: Caroline Sabin (Director), Anthony Nardone (PHE lead), Catherine Mercer, Gwenda Hughes, Greta Rait, Jackie Cassell, William Rosenberg, Tim Rhodes, KP and Samreen Ijaz.

**Contributors** AG developed the discussion guide, scheduled and conducted the interviews, analysed the transcripts and wrote the manuscript. JF and KP conceived the study. JF recruited patients for interview and provided feedback on the discussion guide and early drafts of the manuscript. MG provided social science advice regarding the analysis, interpretation and reporting of findings, double-coded a subset of the transcripts and revised early drafts of the manuscript. SF, NN, AC, RG and CO organised and carried out recruitment of participants and provided feedback on manuscript drafts. SC provided guidance on interpretation of preliminary findings and feedback on manuscript drafts. KP oversaw the development of fieldwork materials and early revisions of the manuscript. GH provided senior social science guidance on fieldwork, analysis, data interpretation and reporting and revisions to early drafts of the manuscript. All authors contributed to and approved the current version of the manuscript.

**Funding** The research was funded by the National Institute for Health Research Health Protection Research Unit in Blood Borne and Sexually Transmitted Infections at University College London in partnership with Public Health England and in collaboration with the London School of Hygiene and Tropical Medicine. This report is independent research by the National Institute for Health Research. The views expressed in this publication are those of the author(s) and not necessarily those of the National Health Service, the National Institute for Health Research, the Department of Health or Public Health England.

**Competing interests** JF has received grant support from Gilead Sciences and Merck. AG and KP have served on advisory boards for ViiV Healthcare. NN received honoraria for speaking and advising, travel grants and an institutional research grant from Bristol-Myers-Squibb, Janssen, ViiV Healthcare and Gilead Sciences. CO has received research grants, personal fees and non-financial support from Gilead Sciences, ViiV Healthcare, Bristol-Myers Squibb and MSD and Janssen for advisory boards, lectureships and travel. All other authors declare that they have no conflict of interest.

**Patient consent** Patient consent forms were signed by every participating patient. These forms were per protocol, following a standardised format, as a substudy to the UK Register of HIV Seroconverters cohort. The ethics committee and individual R and D departments at each participating NHS Trust approved the consent forms and patient information sheets as part of the ethical approval process.

**Ethics approval** NRES Committee West Midlands-South Birmingham.

**Provenance and peer review** Not commissioned; externally peer reviewed.

**Data sharing statement** No additional data are available.

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
