## [Reviewer comments · BMJ Open]

ARTICLE DETAILS

TITLE (PROVISIONAL)	A qualitative study exploring the social and environmental context of recently-acquired HIV infection among men who have sex with men in South-East England
AUTHORS	Gourlay, Annabelle; Fox, Julie; Gafos, Mitzy; Fidler, Sarah; Nwokolo, Nneka; Clarke, Amanda; Gilson, Richard; Orkin, Chloe; Collins, Simon; Porter, Kholoud; Hart, Graham

VERSION 1 - REVIEW

REVIEWER	Brian Adams University of Pittsburgh, Pittsburgh, Pennsylvania, United States of America
REVIEW RETURNED	02-May-2017

GENERAL COMMENTS	Thank you for this manuscript and for looking into the contextual factors behind HIV transmission. I found the paper very interesting. I think the results section could use a bit of revamping; there is an over-reliance on illustrative quotes, to the point where even the salient quotes lose meaning. For example, on page 10, there are 3 quotes in a row - starting at line 16 - discussing psychological factors that are not linked back to HIV or risk at all. They are also qualified as pertaining to "some" or "several" participants. As a reader interested in the findings of your study, I want to know about the themes that emerged from the interviews and how these themes relate to HIV infection. I am not saying that the early drug and alcohol habits do not pertain to HIV infection necessarily, but the way they are currently presented does not draw any link, as if it is just a report back of the findings of the interviews rather than a synthesis of information. I loved the quote starting with line 5 on page 15. I think quotes like that truly highlight what you are trying to explain, that there are so many external factors that come into play when an individual chooses or doesn't choose to engage in high risk sexual behavior. If I were to organize the "individual and interpersonal" section, I would give less time to each specific individual and interpersonal factor, especially if only a small subset of subjects mentioned them, and focus much more on how those men dealt with the individual and interpersonal factors. Use Figure 1 as a guide; there are 4 individual/interpersonal psycho-social factors listed, and I would explain each one in the context provided to you in the interviews that led you to identify that as an important factor, and then link it back to how it impacts HIV risk. I am confused about the purpose of Figure 2. There is not much explanation of it in the text. What is it about these three examples that is necessary for the reader to understand? Were these the three primary examples that emerged from the interviews? Are they just three examples of how the factors can influence HIV risk and
--

	transmission? I very much enjoyed the synthesis in the discussion section. I believe with a restructure of the results section, this would be a very strong paper.
--	---

REVIEWER	Joseph Perazzo, PhD, RN Case Western Reserve University, USA
REVIEW RETURNED	03-May-2017

GENERAL COMMENTS	Thank you for the opportunity to review this article. The authors are addressing the important issue of environmental and psychosocial factors that contribute to new HIV infections. The authors used qualitative methods to elicit data from recently diagnosed individuals. The following critique includes an overall evaluation of the strengths of the article as well as potential areas of improvement. All critique is given in a spirit of collegiality and helpfulness and never intended to discourage or criticize the work of the authors. Overall Evaluation: This paper is very well written, and provides incredibly rich data in an often difficult-to-reach population. The authors present a strong argument for the need for qualitative research in this area and provide very straight-forward exemplars for each topic they discussed. As detailed below, I would recommend that the authors place a bit more detail in their methods section as well as make the results more structured and succinct. My overall impression of the paper is that the authors were clear in their purpose and present very important work in this piece. Background: There is an evidently strong case for this paper and the authors make clear that there is a notable gap in current knowledge. The authors discuss current statistics regarding condomless sex and the relative paucity of information about specific (non-abstract) factors that are associated with increased risk for these behaviors. I have little critique of their background argument. Methods: The authors drew upon an existing cohort sample and adequately discuss the inclusion criteria from that study. There is no mention of exclusion criteria applied for this study. For example, in most cases it must be verified that the individual has the mental capacity to participate in an interview- were there any criteria that excluded participants? If so, please elaborate in your narrative. In the procedural section, the authors discuss that the interviews were conducted by a "female researcher". I would recommend deleting this detail or elaborating on it- is there significance to the researcher's gender in this case? The greatest critique I have for the methods section is the lack of an established method. In-depth interviewing describes a procedural approach to qualitative research. I would recommend the authors reference an established method that they followed and cite their source. For example, the parsimonious and plain language (e.g. non-theoretical; non-abstract) language suggests the method to be a "qualitative descriptive study guided by socioecological theory" (See
---

Sandelowski 2000's article on qualitative description.)

I would recommend that the authors add the following information that is common in qualitative research: (1) discuss procedures for achieving consensus on codes and how discrepancies etc. were handled (often through thorough discussion); (2) a sentence regarding identification of case exemplars to present to the reader (e.g. quotes that best described the themes); (3) Discuss any actions taken by the research team to verify findings with participants (e.g. member checking)

Results:

The results are rich and in-depth. There are several recommendations I have for improvement to this section. First, in general, if the quote is less than 30 words it should be part of the narrative and not in blocked format. Currently, all quotes (long and short) are in blocked format.

The paper (while superbly written) requires the reader to discover specific themes and sub-themes as they read. The authors will add clarity and structure to the paper by introducing the overall scope of their results at the beginning of this section, and then discuss each section. I can see several avenues the authors could take to achieve this. One possible avenue would be to elevate/cluster the themes to a higher level of abstraction. For example (and only an example), a THEME could be "Psychological Distress" while sub-themes could be (1) Past trauma (2) Recent Trauma. Another avenue would be to present themes that are congruent with the theoretical framework (e.g. PERSONAL FACTORS) and then sub-themes that fit into that theme; another for ENVIRONMENTAL FACTORS and so on. To organize the information I recommend the following procedure:

- (1) Present the main themes at the beginning of the results section (e.g. Participants described personal, environmental, social, and structural factors that influenced their.....)
- (2) At the beginning of discussion of each theme, define the theme and then present the sub themes (Personal factors are ABC (definition)...Our analysis revealed that XYZ (sub-themes) were commonly discussed among participants.
- (3) Present your findings and case exemplars.

The quotes provided are quite good. There are conflicting opinions about the best way to use quotations in scholarly writing. Unfortunately, quotations make the paper much longer and, sadly, their rich content often must be cut. You will save a lot of space by integrating short quotes into the narrative paragraphs. However, more importantly, do not be afraid to elaborate more of your team's analysis of the data- saving only those quotations with the most impact for the narrative. Some of the quotations do not appear to be significantly different than previously presented quotations, which truly detracts from their impact. I would recommend the authors revise this section to include their analytic presentation of the meaning behind some of the quotations in each section. I completely understand that this is very detailed and meticulous analysis, and we want to give as much of a voice to the participants as possible- however, don't be afraid to assert more of your analysis. For example, several quotes about abuse, violence, and recent stress could potentially be removed, and you can discuss your overall findings.

Ultimately the number of quotations is between the authors and the

	editorial team- this is merely a suggestion. A method I have used is to create a supplemental two-column table in which I add the theme (LEFT COLUMN) and insert as many quotes as I can find (RIGHT COLUMN). This is a simple way to make your data and analysis transparent to the reader, and show the frequency with which these items were discussed across interviews (the quantity is not necessarily important, but it allows you to present the content). Forgive me- I may just be unaware, but in one quotation the participant states that he “went under on G”; please add a bracket to explain what this means. Discussion: As with the background, I have very little to critique in this section. The authors do a fine job of reiterating their results (should you choose to take my advice above, I would reiterate themes and sub-themes). The authors relate their findings to previous research well but also demonstrate how their work adds to the current state of the science. Limitations and directions for future inquiry are sound. Thank you again for the opportunity to review this paper.
--	---

REVIEWER	Yu Cheng Zhongshan School of Medicine, Sun Yat-sen University, China
REVIEW RETURNED	14-May-2017

GENERAL COMMENTS	The paper aims to identify the individual psycho-social factors and the factors in broader social context influencing risk behaviours and HIV acquisition and reveal the complex interplays between these two types of factors by conducting qualitative research. One advantage is that the paper is driven by existing theories which have been presented in the background section; the other advantage of the paper is well-designed research method. At present, however, it still has theoretical limitations. It would become publishable if the authors can deal with the following issues. 1) As the authors state in the Discussion section, ‘a complex web of factors operating at different level’ (P27 Line 33) and ‘revealed a complex interplay of factors at the level of individuals and their community’ (P24 Line 7). However, what I can read is that the factors at individual and interpersonal level and community and structural level did influence HIV acquisition significantly, but it is not at all surprising because these research findings echo existing research. The paper hasn’t presented a strong link between the factors at both levels, i.e. how these factors interact with each other crossing different levels. 2) Accordingly, the empirical data supporting the augment on ‘complex interplay’ is inadequate (P20-23). 3) In terms of the use of empirical date, the paper lacks necessary explanation why specific piece of date is relevant to the statements and issues. For example, the authors use ‘I grew up in a hostile environment...a small village [abroad], and being gay is not what parents wish for their kids...If you are gay it’s quite difficult for yourself’ to support the argument of ‘A few men were raised in
---

	homophobic or highly religious cultures, mostly abroad and/or in rural settings, which shaped self-perceptions regarding sexual identity.' (P10-11) The author should have explained why this issue is relevant to HIV acquisition clearly. Besides, 'being gay is not what parents wish for their kids' is still quite normal in many cultures or countries. Therefore, I suggest the authors to rethink all the arguments from P9-23 where they tried to theorize their points of view based on the empirical data.
--	--

VERSION 1 – AUTHOR RESPONSE

Reviewer: 1

Reviewer Name
 Brian Adams

Institution and Country
 University of Pittsburgh, Pittsburgh, Pennsylvania, United States of America

Please state any competing interests or state 'None declared':
 None declared

Please leave your comments for the authors below Thank you for this manuscript and for looking into the contextual factors behind HIV transmission. I found the paper very interesting. I think the results section could use a bit of revamping; there is an over-reliance on illustrative quotes, to the point where even the salient quotes lose meaning. For example, on page 10, there are 3 quotes in a row - starting at line 16 - discussing psychological factors that are not linked back to HIV or risk at all. They are also qualified as pertaining to "some" or "several" participants. As a reader interested in the findings of your study, I want to know about the themes that emerged from the interviews and how these themes relate to HIV infection. I am not saying that the early drug and alcohol habits do not pertain to HIV infection necessarily, but the way they are currently presented does not draw any link, as if it is just a report back of the findings of the interviews rather than a synthesis of information. I loved the quote starting with line 5 on page 15. I think quotes like that truly highlight what you are trying to explain, that there are so many external factors that come into play when an individual chooses or doesn't choose to engage in high risk sexual behavior. If I were to organize the "individual and interpersonal" section, I would give less time to each specific individual and interpersonal factor, especially if only a small subset of subjects mentioned them, and focus much more on how those men dealt with the individual and interpersonal factors. Use Figure 1 as a guide; there are 4 individual/interpersonal psycho-social factors listed, and I would explain each one in the context provided to you in the interviews that led you to identify that as an important factor, and then link it back to how it impacts HIV risk.

We have reviewed the quotes and removed those considered to illustrate relatively more minor points. Regarding the linking of quotes and themes to HIV acquisition, we intended the first three results sub-sections within the "individual and interpersonal" section to highlight the factors that could lead to vulnerabilities and impacts on psychological wellbeing, while the final sub-section demonstrates how these intermediary outcomes can lead, in turn, to risk situations including risk of HIV infection. We focus further on links to HIV acquisition in the final results section "combined effects at the interface of individual and community levels on risk". In order to make this clearer, and in view of suggestions made by reviewer 2 (below), we have now added an introductory paragraph to the beginning of each section within the results (page 9; page 15 and page 20). These extra paragraphs provide an overview of the themes identified at each level, e.g. the 4 individual/interpersonal psycho-social factors listed in figure 1, and explain the structuring, including sign-posting to where the links to risk

behaviours and risk of HIV infection are described.

I am confused about the purpose of Figure 2. There is not much explanation of it in the text. What is it about these three examples that is necessary for the reader to understand? Were these the three primary examples that emerged from the interviews? Are they just three examples of how the factors can influence HIV risk and transmission?

The purpose of figure 2 is primarily to highlight the complexity, showing how individuals experienced a combination of factors from different levels that influenced their risk of HIV infection, but also how the balance and importance of factors from each level differed for each individual. Contrasting cases were selected, with case A illustrating numerous contributing psycho-social factors at the individual level, which had a greater bearing on risk behaviour and risk of HIV acquisition than community/structural-level factors. Case B, while also illustrating a greater influence of psycho-social factors than community/structural-level factors, shows how more recent psycho-social stressors were important, in contrast to case A, where psycho-social factors were experienced cumulatively since childhood – themes that were highlighted in the results. Case C contrasts with cases A and B, in that community/structural-level factors were relatively more important in contributing to HIV risk than the more limited set of psycho-social factors experienced. A brief explanation was originally included in the footnote to figure 2 (page 33), but we have now added further explanation in the text to emphasize the purpose of this figure and clarify it for the reader (page 24).

I very much enjoyed the synthesis in the discussion section. I believe with a restructure of the results section, this would be a very strong paper.

Reviewer: 2

Reviewer Name

Joseph Perazzo, PhD, RN

Institution and Country

Case Western Reserve University, USA

Please state any competing interests or state 'None declared':

None Declared

Please leave your comments for the authors below Thank you for the opportunity to review this article. The authors are addressing the important issue of environmental and psychosocial factors that contribute to new HIV infections. The authors used qualitative methods to elicit data from recently diagnosed individuals. The following critique includes an overall evaluation of the strengths of the article as well as potential areas of improvement. All critique is given in a spirit of collegiality and helpfulness and never intended to discourage or criticize the work of the authors.

Overall Evaluation:

This paper is very well written, and provides incredibly rich data in an often difficult-to-reach population. The authors present a strong argument for the need for qualitative research in this area and provide very straight-forward exemplars for each topic they discussed. As detailed below, I would recommend that the authors place a bit more detail in their methods section as well as make the results more structured and succinct. My overall impression of the paper is that the authors were clear in their purpose and present very important work in this piece.

Background:

There is an evidently strong case for this paper and the authors make clear that there is a notable gap in current knowledge. The authors discuss current statistics regarding condomless sex and the

relative paucity of information about specific (non-abstract) factors that are associated with increased risk for these behaviors. I have little critique of their background argument.

Methods:

The authors drew upon an existing cohort sample and adequately discuss the inclusion criteria from that study. There is no mention of exclusion criteria applied for this study. For example, in most cases it must be verified that the individual has the mental capacity to participate in an interview- were there any criteria that excluded participants? If so, please elaborate in your narrative.

We indicated the inclusion criteria for the qualitative sub-study in paragraph 3 on page 6, i.e. eligible for the UK Register cohort study (inclusion criteria detailed in paragraph 1, page 6), aged ≥ 18 years, diagnosed within the last 12 months, and attending one of six centres in London/Brighton. As participants were recruited via the UK Register cohort study, there were no further exclusion criteria, other than participants being willing to undertake the interview and give their consent, which has now been clarified in the manuscript (paragraph 3, page 6).

In the procedural section, the authors discuss that the interviews were conducted by a “female researcher”. I would recommend deleting this detail or elaborating on it- is there significance to the researcher’s gender in this case?

We agree that this detail can be deleted and have done so accordingly. We had not discussed how the gender of the interviewer may/may not influence participants’ responses, as we prioritised discussion around other methodological aspects and aimed to keep the paper succinct as possible.

The greatest critique I have for the methods section is the lack of an established method. In-depth interviewing describes a procedural approach to qualitative research. I would recommend the authors reference an established method that they followed and cite their source. For example, the parsimonious and plain language (e.g. non-theoretical; non-abstract) language suggests the method to be a “qualitative descriptive study guided by socioecological theory” (See Sandelowski 2000’s article on qualitative description.)

We have added a sentence describing the overall study method (paragraph 2, page 6) which we agree is best described as a qualitative descriptive study. We have also included two references. Paragraph 1, page 8 describes in further detail the analysis methods used, including references.

I would recommend that the authors add the following information that is common in qualitative research: (1) discuss procedures for achieving consensus on codes and how discrepancies etc. were handled (often through thorough discussion); (2) a sentence regarding identification of case exemplars to present to the reader (e.g. quotes that best described the themes); (3) Discuss any actions taken by the research team to verify findings with participants (e.g. member checking) We compared coding and themes identified through the comparison and discussion of thematic maps (paragraph 1, page 8), and handled any discrepancies through discussion, which we have clarified in the text on page 8. We have also added a sentence describing how quotes were selected (paragraph 1, page 8). We were not able to verify findings with participants, e.g. through member checking, for logistical reasons and given participants were only consented to one interview.

Results:

The results are rich and in-depth. There are several recommendations I have for improvement to this section. First, in general, if the quote is less than 30 words it should be part of the narrative and not in blocked format. Currently, all quotes (long and short) are in blocked format.

We considered this suggestion but found that integrating shorter quotes into the text did not necessarily save words, due to the need to add further introductory text, and also made the quotes harder to identify. We have therefore opted to retain all quotes in their original indented format, which appears to conform to the style of the journal.

The paper (while superbly written) requires the reader to discover specific themes and sub-themes as they read. The authors will add clarity and structure to the paper by introducing the overall scope of their results at the beginning of this section, and then discuss each section. I can see several avenues the authors could take to achieve this. One possible avenue would be to elevate/cluster the themes to a higher level of abstraction. For example (and only an example), a THEME could be "Psychological Distress" while sub-themes could be (1) Past trauma (2) Recent Trauma. Another avenue would be to present themes that are congruent with the theoretical framework (e.g. PERSONAL FACTORS) and then sub-themes that fit into that theme; another for ENVIRONMENTAL FACTORS and so on. To organize the information I recommend the following procedure:

- (1) Present the main themes at the beginning of the results section (e.g. Participants described personal, environmental, social, and structural factors that influenced their.....)
- (2) At the beginning of discussion of each theme, define the theme and then present the sub themes (Personal factors are ABC (definition)....Our analysis revealed that XYZ (sub-themes) were commonly discussed among participants.
- (3) Present your findings and case exemplars.

We agree with the reviewer that adding further discussion of the scope of each results section would help add clarity to the results. We presented the overarching structuring of themes at the personal, community and structural levels at the beginning of the results (paragraph 2, page 9) and have now added introductory paragraphs at the beginning of each section to provide an overview of the themes that were identified (page 9, page 15 and page 20).

The quotes provided are quite good. There are conflicting opinions about the best way to use quotations in scholarly writing. Unfortunately, quotations make the paper much longer and, sadly, their rich content often must be cut. You will save a lot of space by integrating short quotes into the narrative paragraphs.

As indicated above, we found that integrating shorter quotes into the text did not necessarily reduce the length/word count, so opted to retain the indented format for clarity and consistency.

However, more importantly, do not be afraid to elaborate more of your team's analysis of the data- saving only those quotations with the most impact for the narrative. Some of the quotations do not appear to be significantly different than previously presented quotations, which truly detracts from their impact. I would recommend the authors revise this section to include their analytic presentation of the meaning behind some of the quotations in each section. I completely understand that this is very detailed and meticulous analysis, and we want to give as much of a voice to the participants as possible- however, don't be afraid to assert more of your analysis. For example, several quotes about abuse, violence, and recent stress could potentially be removed, and you can discuss your overall findings.

We have reviewed the included quotes and removed those that do not further enrich the findings or are similar to others already shown. Where applicable, we have added some extra description in the text instead.

Ultimately the number of quotations is between the authors and the editorial team- this is merely a suggestion. A method I have used is to create a supplemental two-column table in which I add the theme (LEFT COLUMN) and insert as many quotes as I can find (RIGHT COLUMN). This is a simple way to make your data and analysis transparent to the reader, and show the frequency with which these items were discussed across interviews (the quantity is not necessarily important, but it allows you to present the content).

We thank the reviewer for their suggested approach, but do not feel it is necessary in this case to include a lengthy supplemental table with all the possible quotes for each theme, as we have endeavoured to select quotes for the reader that best illustrate the themes.

Forgive me- I may just be unaware, but in one quotation the participant states that he "went under on

G"; please add a bracket to explain what this means.

We have clarified in brackets (page 17) that this participant is referring to being under the influence of GHB/GBL.

Discussion:

As with the background, I have very little to critique in this section. The authors do a fine job of reiterating their results (should you choose to take my advice above, I would reiterate themes and sub-themes). The authors relate their findings to previous research well but also demonstrate how their work adds to the current state of the science. Limitations and directions for future inquiry are sound.

Thank you again for the opportunity to review this paper.

Reviewer: 3

Reviewer Name

Yu Cheng

Institution and Country

Zhongshan School of Medicine,

Sun Yat-sen University,

China

Please state any competing interests or state 'None declared':

None declared

Please leave your comments for the authors below The paper aims to identify the individual psycho-social factors and the factors in broader social context influencing risk behaviours and HIV acquisition and reveal the complex interplays between these two types of factors by conducting qualitative research. One advantage is that the paper is driven by existing theories which have been presented in the background section; the other advantage of the paper is well-designed research method. At present, however, it still has theoretical limitations. It would become publishable if the authors can deal with the following issues.

1) As the authors state in the Discussion section, 'a complex web of factors operating at different level' (P27 Line 33) and 'revealed a complex interplay of factors at the level of individuals and their community' (P24 Line 7). However, what I can read is that the factors at individual and interpersonal level and community and structural level did influence HIV acquisition significantly, but it is not at all surprising because these research findings echo existing research. The paper hasn't presented a strong link between the factors at both levels, i.e. how these factors interact with each other crossing different levels.

We aimed to focus on the complexity and importance of combinations of factors from each level in the final sub-section of the results "combined effects at the interface of individual and community levels on risk", highlighting that most participants thought their HIV infection was a result of a combination of multiple factors within and across different levels, and that the specific sets of contributory factors differed in relative importance for each individual (page 20). We also endeavoured to describe in this section how factors from more than one level could come together and magnify risk, e.g. individual-level psycho-social trauma, in combination with community-level perceptions of HIV, altered attitudes to risk and decisions around condomless sex. We have now added an introductory paragraph to this section of the results, following from reviewer 2's feedback above, which we believe strengthens this section by summarising the ways in which factors at each level came together to influence risk. We

have also added sub-section headings in this section for additional clarity.

Figure two also aimed to consolidate the findings reported in the text, demonstrating the complexity and importance of combinations of factors from each level, as well as contrasting sets of influences for different individuals. In light of a comment by reviewer 1 regarding figure 2, we have now strengthened our description of figure 2 in the text (page 24), to consolidate the messages and further highlight the complexity of factors.

2) Accordingly, the empirical data supporting the argument on 'complex interplay' is inadequate (P20-23).

We have reviewed all quotes in this section and feel that the empirical data presented do support a layered complexity. For example, the following quote (page 23) demonstrates how psycho-social factors at the individual level, e.g. psychological trauma, as well as community/structural-level factors, e.g. changing community perceptions of HIV in light of the availability of ART, together altered perceptions of risk and HIV acquisition:

When we were young adults, the fear of God was put into us. If you got it [HIV] you died...and suddenly you get to a point now where it is manageable...you could live a normal life...and you are thinking 'Okay, I live in a city with very high prevalence of this disease. I am at a crossroads in my life in terms of what I want to experience sexually'...I think the trauma that I have gone through has changed what I perceive as real risk in life...There are far more important things...So it was a combination of all those different aspects, I had actually come to the conclusion that if I did become HIV-positive it wouldn't be a big event in my life.

The magnified effects and importance of combinations of mutually exacerbating factors, as illustrated in the selected quotes, is further supported by syndemic theory, as discussed on page 25.

3) In terms of the use of empirical data, the paper lacks necessary explanation why specific piece of data is relevant to the statements and issues. For example, the authors use 'I grew up in a hostile environment...a small village [abroad], and being gay is not what parents wish for their kids...If you are gay it's quite difficult for yourself' to support the argument of 'A few men were raised in homophobic or highly religious cultures, mostly abroad and/or in rural settings, which shaped self-perceptions regarding sexual identity.' (P10-11) The author should have explained why this issue is relevant to HIV acquisition clearly. Besides, 'being gay is not what parents wish for their kids' is still quite normal in many cultures or countries. Therefore, I suggest the authors to rethink all the arguments from P9-23 where they tried to theorize their points of view based on the empirical data.

We have reviewed all the quotes, checked that they support the points being made, and removed some quotes where appropriate. The particular quote mentioned by the reviewer has been replaced with a different example which links more closely with the text, with further description of the issue/sub-theme also having been added (page 11).

Further to the comments from reviewers 1 and 2, we have now added an introductory paragraph to the beginning of each sub-section in the results (page 9, page 15 and page 20) to clarify the structuring, sign-posting to where the links to risk behaviours and risk of HIV infection are described. We were not attempting to demonstrate causality, but rather the layering of factors that build up to risk situations and the potential for HIV infection. As such, the "individual and interpersonal" results section highlights the factors that can lead to vulnerabilities and impacts on psychological wellbeing, while the final sub-section "impacts on risk behaviour" demonstrates how these intermediary outcomes can lead, in turn, to risk situations including the potential risk of HIV infection. We focussed further on links to HIV acquisition in the final results section "combined effects at the interface of individual and community levels on risk". We have now strengthened this section through the addition of a new initial paragraph (page 20) which summarises how factors from each level can operate in

combination and interact to affect HIV risk, as well as consolidating the description of figure 2 (page 24).

VERSION 2 – REVIEW

REVIEWER	Brian Adams University of Pittsburgh, Pittsburgh, PA, United States
REVIEW RETURNED	05-Jul-2017

GENERAL COMMENTS	This revision is excellent. My primary concerns with the original version came in the results section, for which I suggested a restructure of the section and a reduction in the number of quotes in order to provide the quotes with more impact. This revision addresses both of these concerns. The addition of the titles for later results sections and the general descriptions for each section helped to focus me as a reader. The quotes used were not only relevant, but were also accompanied by author interpretation. I also appreciated the expanded explanation of Figure 2, and it now not only makes sense but drives home a very important point in the manuscript. As I predicted in my first review, the restructure of the results section turned this into a very strong paper. Well done.
--

REVIEWER	Joe Perazzo, PhD, RN Case Western Reserve University; Cleveland, OH USA
REVIEW RETURNED	29-Jun-2017

GENERAL COMMENTS	I am very impressed with this revision- the authors made all suggested revisions that I provided in the first read. The results section, which was already very good in my opinion, now reads a bit more easily with the structure the authors added. Furthermore they added enriched content with their description of each theme. Well done. I do respect the authors' decision not to integrate short quotations into the narrative- but would like to state that in reporting such findings they should be in quotes or italicized (so they are not hard to distinguish), and the suggestion saves space rather than words. However, this is more of an editorial preference and if the editors do not require it I certainly yield to the preferences of the authors. The supplementary file was merely a suggestion- nothing that would stop the paper from publishing. I agree that the paper is not lacking in detail to support the themes. I do hope the authors will consider this action in future work... as this sort of data transparency is very important to scientific discourse, but is not often practiced in qualitative studies. I have little other critique- and it appears that they have addressed all of the critiques that were asked of them (by me and the other reviewers). thank you for the opportunity to review this paper. It covers a VERY important topic- and the authors asked "hard" questions, and
---

	produced valuable data as a result. Great job from my perspective.
--	--